## Research Article

humanitarian; person-centred care; quality of care; refugees; responsiveness

**Corresponding author:**
Michael McGrath;
Email: michael.mcgrath@unsw.edu.au

# Contextualising person-centred mental health and psychosocial support (MHPSS) services: A qualitative study of the preferences and experiences of displaced Syrians in Northwest Syria and Türkiye

Michael McGrath[1] [iD], Wael Yasaki[2], Ammar Beetar[2], Ahmed El-Vecih[2], Louis Klein[1], Gulsah Kurt[1] [iD], Salah Lekkeh[2], Simon Rosenbaum[1] and Ruth Wells[1]

[1]Discipline of Psychiatry and Mental Health, University of New South Wales, Sydney, Australia and [2]Hope Revival Organization, Gaziantep, Türkiye

## Abstract

Understanding and responding to patient expectations is crucial for providing high-quality, person-centred mental healthcare, but remains underexplored in humanitarian settings. This study examines the preferences and experiences of Syrian mental health and psychosocial support (MHPSS) service users in Northwest Syria and Türkiye. We conducted structured interviews with 378 displaced Syrians (55% female, mean age: 31 years). Participants completed the Client Satisfaction Questionnaire-8 and responded to nine open-ended questions. An abductive qualitative content analysis guided by the World Health Organization's health system responsiveness framework was used to interpret their accounts. Participants most frequently described the importance of time and understanding (62%), dignity (43%), confidentiality (36%) and continuity of care (31%), with notable variation by gender. Interpersonal aspects of care were crucial for building trust and sustaining service engagement. Service-level factors, such as adequate time with practitioners and integrated and coordinated care, ensured high-quality support in a context of ongoing conflict, displacement and poverty. These findings underscore the importance of embedding person-centred approaches in MHPSS service design and delivery. As efforts to rebuild Syria's health system begin, prioritising service user experiences could improve the quality of care and restore health system trust and legitimacy.

## Impact statement

Efforts to close the global mental health "treatment gap" and achieve universal health coverage require not only the expansion of services but also improved quality of care. This study highlights person-centredness as a critical but overlooked dimension of quality in humanitarian settings. Drawing on structured interviews with displaced and conflict-affected Syrian mental health and psychosocial support (MHPSS) service users, the research identifies what matters most to people receiving care. Participants emphasised the interpersonal aspects of care, such as dignity, understanding and autonomy, as well as structural factors like adequate time, continuity of provider and integrated and coordinated services. These were not peripheral concerns but were central to service satisfaction and ongoing engagement. Our findings have practical implications for the design and delivery of MHPSS services in humanitarian settings. They demonstrate the importance of listening to service users and ensuring care is aligned with their needs and priorities. Our findings could inform efforts to rebuild Syria's health system, which has been affected by decades of conflict and authoritarianism. Sustained investment in the mental health workforce and locally led organisations, best placed to provide person-centred care, could rebuild health system confidence and legitimacy and promote long-term engagement with care.





## Introduction

Recent global health initiatives have prioritised improving quality of care in low- and middle-income countries, reflecting growing recognition that achieving universal health care and closing the mental health treatment gap requires both the expansion of service coverage and improved quality of care (Kruk et al., 2018; National Academies of Sciences, 2018; Patel et al., 2018). Person-centredness is a core component of quality of care, alongside effectiveness, safety, equity, timeliness and efficiency (WHO, 2018a), and is defined as care that is "respectful of and responsive to individual preferences, needs and values" (National Academies of Sciences, 2018).

Providing person-centred care builds trust and confidence in health systems, increases service engagement and improves health outcomes (Doyle et al., 2013), while ensuring individuals' values are respected and their right to dignified care is upheld (Gostin et al., 2003). However, globally, many people continue to report inattentive or disrespectful treatment, insufficient time with providers, poor communication and long wait times (Kruk et al., 2018). Health systems often remain difficult to navigate and resistant to shared decision-making, leading to calls to re-orient mental health care towards approaches that prioritise individuals' goals and embed dignity and collaboration into service delivery (Davidson and Tondora, 2022; Kruk et al., 2018).

The World Health Organization's (WHO) concept of responsiveness is one of the most widely used frameworks for measuring patient experience (WHO, 2000). It refers to how well health systems meet people's expectations and is recognised as a core health system goal, ensuring that care is respectful and aligned with individuals' needs and right to autonomy. The framework identifies seven domains: dignity, autonomy, confidentiality, prompt attention, quality of basic amenities, choice of provider and clear communication. Responsiveness is widely used to assess experiential quality of care, capturing both patient–provider interactions and service-level aspects of care (Fifield et al., 2022; Kim et al., 2021; Ratcliffe et al., 2020). Despite its importance, responsiveness remains one of the most underexplored aspects of health system strengthening (Ratcliffe et al., 2020), with limited evidence from conflict-affected humanitarian settings (Khan et al., 2021). As a dynamic process shaped by individuals' health needs, outcomes and contexts, there is a need for a deeper understanding of the factors that influence people's experiences of care (Mirzoev and Kane, 2017) and for context-specific research that empirically tests existing frameworks and their relevance across health systems and settings (Khan et al., 2021).

In humanitarian contexts, person-centred care is difficult to prioritise (Al-Jadba et al., 2024; WHO, 2020). Limited resources, an overstretched health workforce, the destruction of infrastructure and a high burden of unmet needs place substantial strain on providers. Instead, the urgency of providing life-saving interventions often takes precedence. The experiences of service users in these settings remain poorly understood, and the meaningful involvement of affected communities is rare (Ansbro et al., 2022; Bogale et al., 2024; Rass et al., 2020). However, neglecting patient preferences can undermine trust, reduce service engagement and represent a poor use of limited resources (WHO, 2020). As humanitarian crises are increasingly protracted, person-centredness must be prioritised, particularly for mental health conditions that require longer-term treatment and coordination (Jordan et al., 2021).

The Syrian Civil War created one of the world's largest humanitarian crises, with 16.7 million people currently in need of assistance (UNHCR, 2025). Exposure to conflict, displacement, loss and other traumatic events, alongside post-displacement stressors, has resulted in high rates of psychological distress (Acarturk et al., 2021; Wells et al., 2016). Mental health and psychosocial support (MHPSS) services take a multi-sectoral approach to protecting and promoting mental health and well-being in humanitarian settings, encompassing basic services, community-based psychosocial support and specialised care (IASC, 2006). However, a substantial treatment gap persists, with many Syrians not seeking care despite experiencing distress (Fuhr et al., 2020). Global efforts to close this gap typically focus on increasing the supply of services rather than fostering demand, neglecting whether available services meet community expectations and preferences (Patel, 2014; Roberts et al.,

2022). In the Syrian context, limited engagement may reflect a failure to respond to the lived realities and needs of a community who have experienced persistent indignity, exclusion and disempowerment (Mansour, 2018). To address this, services must be centred on the perspectives of service users and responsive to their needs and priorities. This study, therefore, aims to (1) explore the preferences and experiences of Syrian MHPSS service users in Northwest Syria and Türkiye and (2) investigate how these elements shape service engagement.

## Methods

### Study design and participants

We conducted a secondary analysis of data collected within the Caring for Carers study, a mixed-methods, quasi-experimental study of a supportive supervision intervention for MHPSS practitioners in Northwest Syria and Türkiye. A full study protocol has been published elsewhere (Wells et al., 2023). Practitioners were recruited to the intervention through the network of project partners (Hope Revival Organisation) in the region. In addition to the supervision intervention, we conducted structured data collection with service users of participating practitioners. Between April 2022 and February 2024, we recruited service users who had attended an MHPSS session with 45 Syrian MHPSS practitioners (30 psychosocial workers, 7 psychologists and 8 protection/MHPSS case workers) working within 9 participating organisations. Eligible participants were Syrian nationals, aged 18 years or older, who attended an MHPSS session in the previous 3 weeks, either as the service user themselves or as an accompanying parent or carer. Ethical approvals were obtained from the University of New South Wales (HC210824) and Koç University (2021.395.IRB.3.182). Verbal informed consent was obtained from all participants, who received no payment.

### Setting

All participants lived in Northwest Syria or Türkiye and attended an MHPSS session with a Syrian practitioner. At the time of our study, Northwest Syria had been operating as a de facto state independent from the Syrian regime since 2013. Around 4.5 million people lived in the region, two-thirds of whom were displaced, and limited humanitarian assistance was available through cross-border assistance from Türkiye. Participants lived in a mix of urban areas, camps and informal settlements. The region lacked a formal health system, and healthcare professionals faced extreme challenges, including the destruction of infrastructure, ongoing conflict, disrupted supply chains and health workforce shortages (Al-Abdulla et al., 2023; Alaref et al., 2023). In Türkiye, over 3.2 million Syrians were registered for temporary protection in 2023 (Göç İdaresi Başkanlığı, 2025). Most live in urban areas, can access some health and social services and receive employment rights. The 2023 Türkiye–Syria earthquake occurred during data collection, further compounding the humanitarian crisis (Jabbour et al., 2023).

### Data collection

Structured telephone interviews were conducted between April 2022 and February 2024 in Arabic by three Syrian research assistants with professional MHPSS experience in the region. Participants completed the Client Satisfaction Questionnaire-8 (CSQ-8) (Attkisson and Zwick, 1982), a validated eight-item tool for assessing patient

satisfaction with health services, using a four-point Likert scale. The questionnaire was translated into Syrian Arabic by the research team (see Supplementary Material). Scores range from 8 to 32, with higher scores indicating greater satisfaction. Seven open-ended follow-up questions were asked during the delivery of the CSQ-8, along with two additional questions about the utility and planned implementation of what was learned during the session.

## Data analysis

Research assistants entered all data into Kobo Toolbox (Pham et al., 2019). Responses to the open-ended questions were translated into English and collated (in both English and Arabic) into transcripts alongside sociodemographic variables and service information. During a process of data familiarisation and discussion within the team, we identified conceptual similarities with the WHO responsiveness framework. We conducted an abductive, qualitative content analysis, which allowed for iteration between data and theory to test and modify the framework in response to new and unexpected findings (Tavory and Timmermans, 2014). A preliminary coding framework included the WHO's seven domains of responsiveness, alongside adaptations to this framework developed in studies of mental health services in Iran (Forouzan et al., 2011) and Germany (Bramesfeld et al., 2007). New codes were developed inductively and reviewed collaboratively to ensure relevance and consistency. Existing elements of the WHO framework were modified to reflect the cultural understanding and contextual realities described by service users. Iteratively blending inductive and deductive coding allowed for emerging findings to be embedded into existing conceptualisations of person-centred care, while expanding the applicability of the framework to this setting (Vila-Henninger et al., 2024).

Two researchers independently coded a subset of transcripts to develop and refine the codebook and assess intercoder agreement. Coding focused on experiential and relational aspects of care, including practitioner–service user relationship, other interactions with the health service, and expressed preferences for care, rather than descriptions of symptom reduction or improved health. One researcher was Australian and coded the English transcripts, while the other, a Syrian researcher with a lived experience of displacement, coded in both languages. To assess consistency and cross-cultural interpretation, inter-rater reliability was measured using Cohen's kappa statistic. Following confirmation of very strong inter-coder reliability ($k = 0.929$), the first researcher coded the remaining transcripts. All interviews were coded to capture perspectives across a range of MHPSS provision, from specialist care to community-based support and protection in both Northwest Syria and Türkiye. This approach followed an information-power rationale, which prioritises contextual relevance and subgroup coverage over numeric thresholds or thematic saturation (Malterud et al., 2016).

Descriptive statistics summarise service user demographics, service utilisation and satisfaction and the responsiveness domain rankings. Blending qualitative analysis and descriptive statistics helped identify aspects of person-centredness that were both numerically and conceptually important, and to understand their relative significance in this setting. Analyses were conducted in NVivo and RStudio.

## Results

Interviews were conducted with 378 participants (55% female, mean age: 31 years) (Table 1). Service users had a median of three

**Table 1.** Participant characteristics

| | |
|---|---|
| Gender (*n*, %) | |
| Male | 169 (44.7) |
| Female | 209 (55.3) |
| Age (mean, range) | 31 (18–66) |
| Marital status[a] (*n*, %) | |
| Married | 233 (75.4) |
| Single | 54 (17.5) |
| Divorced | 12 (3.9) |
| Widowed | 10 (3.2) |
| Household size[a] (mean, range) | 6 (1–16) |
| Previous sessions with this practitioner (median, range) | 3 (0–35) |
| Previous sessions with this service (median, range) | 3 (0–40) |
| Proxy interviews[b] (*n*, %) | 34 (9.0) |
| Client Satisfaction Questionnaire 8 score (mean, range) | 27 (12–32) |

[a]Information on marital status and household size was only collected from 309 participants. Proportions are calculated using this denominator.
[b]Interviews conducted with the parent or carer of the service user.

previous MHPSS sessions and reported generally high satisfaction scores (mean: 27, range: 12–32). In all, 91% of interviews were conducted with the service users themselves.

We identified eight elements of person-centred care from MHPSS service user accounts (Table 2). Four elements from the WHO framework are closely aligned with participant accounts: dignity, confidentiality, clear communication and autonomy. Two elements required contextual adaptation: prompt attention was redefined as time and understanding, while choice was reframed as continuity of care. Notably, the quality of infrastructure was not reported by any participants. Two new elements were developed inductively from service user accounts: coordination and integration, and responsiveness to context. Most frequently reported were time and understanding (62% of participants), followed by dignity (43%), confidentiality (36%), continuity of care (31%) and clear communication (24%) (Figure 1). On average, participants described 2.4 elements per interview, with 89% mentioning one or more. Gender differences were notable: 97% of women identified at least one element (mean = 2.9), compared with 80% of men (mean = 1.6). Time and understanding were most frequently described overall, while confidentiality and continuity of care were more common for women and clear communication was more common among men. Each element is described in detail below.

## Dignity

Respectful and dignified treatment was central to positive experiences. Participants emphasised the importance of being treated with "*respect*" (الاحترام), "*humanity*" (الإنسانية) and in a dignified or gracious manner (معاملة راقية) from polite and courteous staff, who warmly welcomed them. They valued finding a practitioner who "*made me feel seen, values me and listens to me*" and "*is caring, humane, and loves to help.*" Many formed close relationships with their practitioner, who treated them "*like a brother*" or "*like a friend.*" It was important to have their concerns taken seriously, receive non-judgemental care and to be listened to "*without being blamed or ridiculed,*" "*without being lectured*" and "*without being mocked.*"

**Table 2.** Elements of person-centred care described by MHPSS service users

| Concept | Definitions |
| --- | --- |
| 1. Dignity | Staff warmly welcome service users and are polite and well-mannered. |
| | Service users are treated with respect and humanity. |
| | Service user concerns are taken seriously; they are listened to without being mocked. |
| | The service is non-discriminatory. It is an accepting place where all are welcome. |
| 2. Confidentiality | Service users can speak freely and openly. Everything said remains confidential. |
| | Service users can attend sessions in secrecy and not tell their family or community. |
| 3. Communication | Practitioners provide clear information and advice about symptoms, diagnosis and treatment, and communicate ideas and skills that help with recovery. |
| | Practitioners use supportive and encouraging language. |
| 4. Autonomy | Service users and the practitioner work together. They cooperate to develop a treatment plan. |
| | Service users are presented with options and decide which treatment to follow. |
| 5. Time and understanding | Service users are given enough time to explain the nature of their problems. |
| | Service users feel listened to, heard, and understood, often for the first time. |
| | Practitioners show insightful listening, understanding, and empathy. |
| 6. Continuity of care | Service users recognise that their treatment will take time and multiple visits. |
| | Service users want to visit the same trusted practitioner again. |
| | Service users need consistent access to medications and treatment. |
| 7. Coordination and integration | Service users prefer integrated services, in which MHPSS services are co-located with other health services. |
| | Practitioners connect or refer service users to social and economic support services. |
| | Service users have access to both specialist and general services. |
| 8. Responsive to context | Services recognise and respond to the realities of life in a humanitarian context. |
| | Practitioners are flexible and adapt to meet service users' circumstances. |

Services were described as inclusive, providing non-discriminatory treatment and were noted to "*welcome anyone with a problem who needs help.*" Participants emphasised the compassion of practitioners accommodating their circumstances, with one participant describing, "*the specialist is extremely respectful and understanding of my problems and forgives me if I am late or miss a session.*" Respect was also expressed through affirming language, with "*kind words and*

*warm encouragement*" from practitioners, making service users feel capable and promoting greater engagement.

### Confidentiality

Participants emphasised the importance of being able to "*say what I want and everything in my heart confidentially without anyone spreading my words.*" The therapeutic value of finding a space "*to vent and talk about my feelings confidentially*" provided emotional relief and fostered a sense of safety. Confidentiality was the aspect of care with the most notable difference by gender, mentioned by 59% of women but only 7% of men. Women frequently described mental health stigma, concerns about being called "*crazy*" (مجنونة), "*mentally ill*" (مريضة نفسياً) or "*psychologically mad*" (مجنونة نفسياً) and the "*shame*" (العار) this would bring upon their families.

Due to stigma and cultural norms discouraging the discussion of family problems outside of the household, many attended sessions in secrecy. Multiple women provided accounts of initially attending group mental health or gender-based violence awareness-raising sessions. Once confidentiality was explained and the trustworthiness of the practitioner was established, they requested individual sessions. As one participant described:

> Like most women, I have many marital problems that I can't talk to my family about because it would ruin my marriage… She gave a group session to women in our neighbourhood, which I liked. I asked for an individual session with her to see if it would help, and it did. … I didn't tell anyone that I attend because if they knew, they would think I'm crazy.

### Communication

Service users emphasised the importance of clear communication in understanding their mental health and treatment options. Practitioners were valued not only for using plain language to explain symptoms, diagnoses and mental health risk factors, but also for using "*kind words*" and reassuring language, which fostered emotional safety and validation. By sharing practical advice, coping strategies and skills to support recovery, service users felt equipped to manage their symptoms and actively participate in decision-making about their care. This positive communication promoted trust and confidence, facilitating greater engagement with services. However, failing to communicate all aspects of treatment reduced confidence in services, as one participant reported:

> The practitioner helped me by explaining that the issue is quite normal and a result of the fear I experienced during my accident. …but he didn't inform me that the treatment takes time, and that the medication has side effects, which affected me greatly. … I feel there are other things I could be doing that he didn't mention.

Clear communication was the third most frequently described category among men, who understood positive communication in terms of logical explanations and scientific approaches. They sought out "*knowledgeable experts and useful information*" and practitioners whose "*way of talking is rational, and information is correct.*" One man injured during shelling explained:

> I am always tense and scared of sudden loud noises. …I benefited from talking with him about the future and the situation in the region. He spoke to me logically and trained me on facing my fear of loud noises. I want to continue with this.

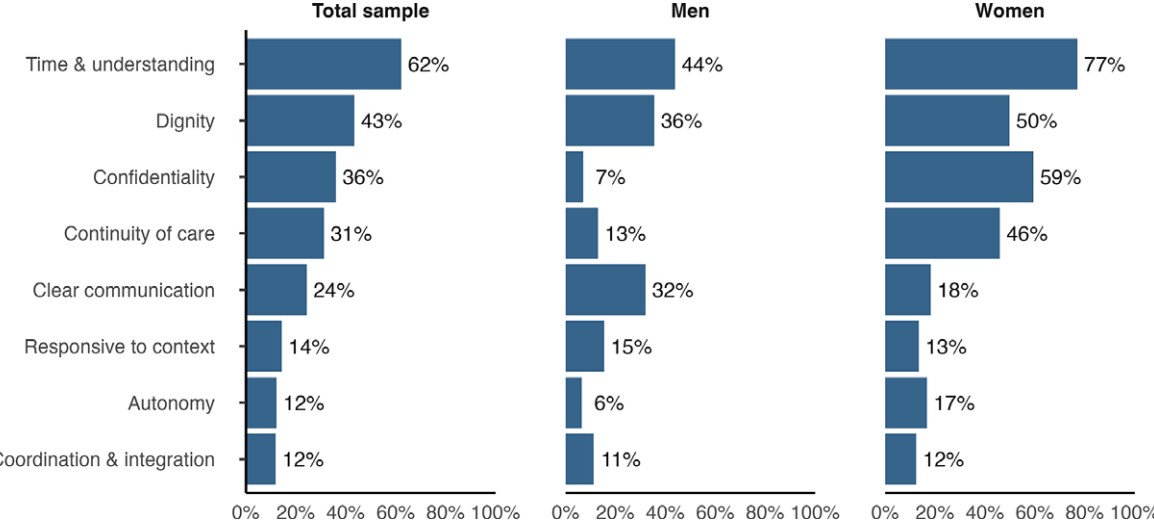

**Figure 1.** Percentage of participants describing each element of person-centred care.

## Autonomy

Service users valued being actively involved in decisions about their care. When practitioners clearly explained treatment options, service users could collaborate in creating and implementing tailored plans. Practitioners and service users worked together to develop goals "*rather than imposing them as advice or orders.*" As one woman described:

> *The specialist never gave me ready-made solutions to my problems. Instead, she presented several solutions and explained the advantages of each one, and I chose the one that suited me and my life best.*

Joint decision-making required not only clear explanations but also active listening and mutual understanding, which reinforced trust and increased engagement with services. One man explained:

> *The specialist cooperates with you in setting treatment goals after listening to you with full attention… You feel understood, heard and appreciated without being blamed or criticised.*

When they were involved in decision-making, service users were more confident in their ability to manage their mental health and respond to future challenges. Service users highlighted the importance of MHPSS services in fostering self-sufficiency and equipping them with meaningful, practical strategies to endure the extreme challenges of life in Northwest Syria. One participant credited her practitioner with fostering resilience and optimism:

> *…she instils strength and hope in me and makes me feel confident about my life. She greatly strengthened my personality. I've come to know how to face the life I'm living.*

## Time and understanding

Rather than describing the need for prompt attention, service users emphasised the importance of quality interactions and dedicated time with attentive practitioners. They valued practitioners who engaged in active listening and demonstrated an understanding of their specific challenges. One participant explained:

> *I need someone to sit with me, understand me, listen to me and give me their time seriously without making fun of me. I need someone to whom I can vent my worries and problems.*

For many, MHPSS sessions were the first time they were able to share their own account of the origin of their distress and discuss their problems openly. They sought practitioners who showed emotional validation and empathy, and tailored their approach to meet individual needs. One woman experiencing social isolation after a divorce noted, "*It's a beautiful thing that there's someone who gives their time to listen to you, share in your emotions, and help you process or transform them.*" Acknowledging the limited support that could be provided in a humanitarian setting, one respondent explained: "*it's enough that there is someone helping you, listening to you and understanding your feelings.*" This aspect of care was particularly important among socially isolated women, many of whom lived with their extended family and had no one they could confide in about their problems. Sharing their concerns was contingent on guaranteed confidentiality from a trusted practitioner. As one woman experiencing marital problems described:

> *I couldn't talk about these things to anyone else, and I felt like I was going to explode… I needed someone to listen to me, understand me, give me useful advice and ideas to help me.*

## Continuity of care

Service users described the need for continuity of care, emphasising the relief of finding a trusted person with whom to discuss their problems over multiple sessions. Rather than emphasising choice of provider, participants described the importance of sustained relationships with their practitioner and building trust over time. A woman supporting her sister who lives with a psychotic disorder recounted:

> *I have been attending the sessions with her for a long time, and my sister has greatly benefited from them. We noticed significant improvements in her thinking and behaviour… She continues to attend the sessions, and we don't want to stop. [The practitioner] has been very good and helpful to us.*

Trust was central to the continuity of care and developed in multiple ways. For many women, trust was dependent on guaranteed confidentiality and the positive experiences of being heard and understood for the first time. One woman experiencing postpartum depression explained:

> *Although I was not convinced anyone could help with my problem… she was very kind and approachable. I trusted her, felt comfortable and began sharing everything about my life, situation and depression.*

*She listened, helped and gave me good ideas, encouraging me to continue attending sessions regularly.*

Less frequently, trust reflected perceptions of practitioner or service quality. This was particularly common among men, who framed high-quality care in terms of scientific or rational approaches and sought out practitioners "*with scientific knowledge who can handle our psychological conditions*" who were "*very knowledgeable about all psychological issues and how to solve them.*"

Participants also recognised that improved mental health would take time and ongoing sessions with their practitioner. A father explained:

*This is our first session, and as a start, it seems good. We need to continue attending sessions to get more help and reach a solution. … There are still many steps and stages towards greater improvement in the coming sessions.*

In Northwest Syria, the conflict limited access to health centres, preventing some from attending sessions, while disruptions to supply chains made it difficult to secure psychotropic medications. In contrast, participants in Türkiye described the relief of having access to medications and services within the Turkish health system. One mother in Türkiye explained:

*My child's mental state is gradually improving, and he is slowly moving away from depression as a result of the psychological support sessions and his adherence to the psychiatric medication prescribed by the doctor.*

### Coordination and integration

Service users explained the value of integrated care, emphasising the importance of accessing multiple forms of support within a single service. Co-locating MHPSS within general health services not only maintained privacy and reduced the stigma of visiting a mental health clinic but also facilitated engagement for those unsure about mental health services or presenting to health centres with somatic symptoms and general health concerns. One participant explained:

*I went to the clinic to have a tooth pulled, and the receptionist noticed I was unhappy and suggested I attend an MHPSS session with the counsellor. Initially, I wasn't convinced that anyone could help me, but she was very warm… I gave her my trust, opened up about my life and depression, and she listened and helped me.*

Participants described the need for services integrating both psychosocial support and specialist mental health services, recognising that they often required multiple levels of care, including referral to psychiatric services. Some described how psychoeducation and brief interventions delivered by psychosocial workers were insufficient to meet their needs. A young man living in a camp described:

*I have a fear of death. I am always afraid of dying and have thoughts about death …I hoped they would treat my emotional problem and help me get rid of my fears. I didn't feel any actual change. The practitioner only taught me deep breathing and relaxation, which is not enough to address my problem.*

Participants valued referral to services that address the problems they understood to be the origin of their distress, such as a lack of food and shelter. A father explained:

*I felt guilt towards my infant daughter because when she was born, I didn't have money for her milk. She got very sick, and I didn't know where to take her, which deteriorated my mental state significantly. The practitioner listened to me, helped me, supported me psychologically, and directed me to a place where they distribute baby milk. …*

*The most important thing is that my daughter has improved due to the practitioner's help.*

Others described the pathway from social and community services into MPHSS services. Without a functioning health system in Northwest Syria, referral processes were often informal, relying on personal networks and the knowledge of practitioners, who went beyond their formal responsibilities to ensure service users could access necessary support.

### Responsive to context

Service users valued care that acknowledged the impact of conflict, displacement and poverty. Many described these factors as the origin of their psychological distress and doubted that MHPSS services alone could alleviate their suffering. As a result, service users valued holistic care from practitioners who were "*considerate of our circumstances*" and provided treatment "*which suits the poor and refugees.*" Practitioners flexibly adapted to the logistical and financial problems faced by service users, overcoming barriers to access caused by the conflict, transport difficulties, physical mobility limitations and poverty. Practitioners instead visited people at home or in their camp, secured free medications, accompanied people with restricted mobility to specialist services and provided care over the telephone. One woman living with a disability explained:

*I cannot always go to the centre, so she gave me her phone number to communicate with her and talk. She listens, advises and helps me a lot with her ideas.*

In Türkiye, receiving care from a Syrian practitioner was especially important. These practitioners had a shared experience of conflict and displacement, provided services in Arabic, understood their culture and faced similar post-migration challenges. These shared experiences fostered trust and strengthened therapeutic relationships. A man in Türkiye observed:

*The specialists are Syrian, from the same culture, and understand the problems we face because they experience similar issues in the refugee community.*

### Discussion

This study examines how service users' experiences of care shape their engagement with MHPSS services in humanitarian contexts. Interpersonal and relational aspects of care were not peripheral considerations but fundamental drivers of service quality and engagement. These findings build on two previous studies adapting the WHO responsiveness framework for mental health care in Iran (Forouzan et al., 2011) and Germany (Bramesfeld et al., 2007), identifying similar expectations around relational aspects of care from mental health services, despite very different populations and settings. Across all three studies, service users prioritised interpersonal relationships and continuity over other domains, such as prompt attention, which was less frequently mentioned. The prominence of these factors may reflect the heightened importance of trust, rapport, therapeutic alliance and emotional safety in mental healthcare, with lessons for strengthening person-centredness in other health settings, including non-communicable diseases (NCDs).

Time and understanding, dignity and confidentiality were the most frequently reported priorities. These aspects of care could be strengthened through greater investment in staff training and

supportive supervision in humanitarian settings, where MHPSS practitioners are often overstretched and manage unrealistic caseloads, limiting the time available for optimal care and comprehensive assessment (Abujaber et al., 2024; Sawah et al., 2025). Humanitarian MHPSS services are often time-limited, leaving practitioners frustrated by needing to abruptly end treatment after a fixed period (Kerbage et al., 2020). These pressures can exacerbate stress and burnout, negatively affecting quality of care, therapeutic relationships and staff retention. Addressing these challenges requires a shift from short-term humanitarian programming towards more sustainable funding and service models.

Our findings contributed to a growing body of literature on continuity of care for mental health (Kerbage et al., 2020) and NCDs (Akik et al., 2024; Ansbro et al., 2021; Arakelyan et al., 2021) in humanitarian contexts. Service users described three interrelated forms of continuity: interpersonal continuity (seeing the same trusted practitioner), longitudinal continuity (across multiple sessions) and management continuity (without interruption) (WHO, 2018b). Interpersonal continuity was shaped by respect, confidentiality, trust and the perceived clinical expertise of the practitioner, while longitudinal and management continuity were dependent on service-level and contextual factors, such as coordination, accessibility and conflict-related disruptions to supply chains. Consistent with other studies, trust was found to be a key determinant of whether service users remained engaged, requiring respect, confidentiality and practitioner attentiveness and understanding (Arakelyan et al., 2021; Noubani et al., 2021). However, MHPSS workers face challenges fostering trust due to high workloads and ambiguity around care coordination (Noubani et al., 2021), while staff shortages and turnover limit opportunities for sustained practitioner–service user relationships (Akik et al., 2024). Notably, women in our study emphasised the importance of confidentiality as a crucial first step in building trust and fostering ongoing engagement, particularly for those seeking support for the first time. In Syria and other Arabic-speaking communities, mental health stigma often reflects worry about family shame, social judgement and reputational harm, particularly for women (Tahir et al., 2022). These concerns may be heightened in camp settings, where a lack of privacy and overcrowded living conditions could intensify concerns about disclosure (Christensen and Ahsan, 2023). These gendered dynamics should be considered when designing and implementing MHPSS services for women.

The need for integrated MHPSS care is well established (IASC, 2006; Tol et al., 2023). Integrating MHPSS services across humanitarian sectors promotes reach, continuity and sustainability, reduces stigma and avoids medicalising social problems (IASC, 2006; Ventevogel, 2014). However, this process of integrating care is poorly understood (Ndlovu et al., 2024). MHPSS services continue to operate separately from primary and NCD care within vertically structured, short-term programmes funded by multiple humanitarian actors with distinct priorities (Gyawali et al., 2021). Our findings confirm the importance of integration for promoting service engagement in settings with substantial mental health stigma and for providing multisectoral support and coordinated care for people with complex needs. Delivering decontextualised services and ignoring the social and economic realities of service users can lead to service disengagement, as stand-alone mental health interventions may seem irrelevant to people whose psychological distress is understood as a response to extreme adversity (Kerbage et al., 2020; Maconick et al., 2020; Roberts et al., 2022). Participants often framed their distress as a direct consequence of displacement, poverty and the recent earthquake, prioritising

support that addressed these hardships. Bidirectional integration of social and health services can avoid duplication and more efficiently use scarce resources to address mental health, NCDs and basic needs concurrently (Gyawali et al., 2021; McNatt et al., 2019; Noubani et al., 2021).

Service user experiences were likely enhanced by the fact that all practitioners in our study were Syrian, living among their community and sharing experiences of displacement, conflict, the earthquake and post-displacement stressors. Shared language and culture are critical for building trust and promoting engagement with mental healthcare, especially in contexts of displacement and trauma (Doğan et al., 2019; Jahan et al., 2024). In a population whose dignity and agency have been undermined by years of exclusion and humiliation, care delivered by fellow Syrians could be a restorative act (Mansour, 2018). As localisation continues to be promoted within the humanitarian system, our findings highlight the importance of local humanitarian responders and refugee-led organisations, in particular their capacity to provide person-centred, context-specific care (Duclos et al., 2021; Singh et al., 2022). However, localisation must be accompanied by sustained investment in local organisations and training and supervision to support local and national mental health professionals, who may face heightened risks of stress, secondary trauma and other adverse mental health outcomes (Cardozo et al., 2005; Strohmeier et al., 2018).

In December 2024, after our data collection had concluded, the Assad regime fell. Most of Syria is now under the control of a transitional government. While the situation remains volatile, attention has shifted to rebuilding Syria's fractured health system. The locally led health service delivery models implemented in opposition-controlled Northwest Syria during the conflict could inform reconstruction efforts (Alkhalil et al., 2025; Marzouk et al., 2025; Nashwan and Swed, 2025). Prioritising service user experiences could help rebuild trust in a health system damaged by decades of conflict and authoritarianism, while ensuring limited resources fund services that reflect patient priorities and needs.

### Limitations

Most participants reported positive experiences and high satisfaction. This may reflect selection bias, as we recruited people currently engaged in services; social desirability bias, leading participants to describe their experiences more favourably; and dependence on services, which may have discouraged open criticism. Data were collected via structured, telephone interviews – the only feasible approach given the ongoing conflict and insecurity. However, this may have limited rapport, depth of disclosure and introduced bias in participation and response compared to face-to-face interviews. Responsiveness reflects normative expectations of health services, shaped by previous experiences and shared values within a "horizon of expectations" (Lakin et al., 2024). These expectations and the relative importance placed on different elements of responsiveness vary across contexts and populations. They are shaped by living conditions, age, gender and past experiences of care (Mirzoev et al., 2025). In humanitarian settings, where health systems have been destroyed by conflict, expectations may be lower, potentially explaining high satisfaction scores and the limited emphasis placed on domains like timeliness or physical infrastructure. Finally, the unique governance and service delivery context of opposition-controlled Northwest Syria may limit the generalisability of these findings to other humanitarian settings.

## Conclusion

Service users emphasised the importance of interpersonal aspects of care, including dignity, autonomy, trust and contextually relevant care, alongside structural factors, such as adequate time, continuity and coordination and integration. These preferences were not secondary concerns, but were rather central to service satisfaction and ongoing engagement. During the rebuilding of Syria's health system, prioritising service user experiences could restore trust, confidence and legitimacy to the health system following decades of conflict and authoritarianism. Building high-quality mental health systems will require long-term investment in the mental health workforce and the local organisations best positioned to deliver person-centred care.

**Open peer review.** To view the open peer review materials for this article, please visit http://doi.org/10.1017/gmh.2025.10058.

**Supplementary material.** The supplementary material for this article can be found at http://doi.org/10.1017/gmh.2025.10058.

**Data availability statement.** Data are not publicly available to protect the privacy of research participants. Data may be available on request from the corresponding author, subject to relevant ethical approvals.

**Author contribution.** MM conceptualised the study, conducted the analysis and wrote the initial draft. MM and WY were responsible for data curation. AEV, SAL, WY, LK and MM managed the data collection process. RW and SR provided supervision. RW, SR and AB acquired the funding. All authors (MM, WY, AB, AEV, LK, GK, SAL, SR and RW) contributed to the interpretation of results, critically reviewed and revised the manuscript and approved the final submission.

**Financial support.** This research is supported by ELRHA's Research for Health in Humanitarian Crises (R2HC) Program (Grant Number: RG203720), which aims to improve health outcomes by strengthening the evidence base for public health interventions in humanitarian crises. R2HC is funded by the UK Foreign, Commonwealth and Development Office (FCDO), Wellcome, and the Department of Health and Social Care (DHSC) through the National Institute for Health Research (NIHR). The funding body had no role in the conceptualisation, writing of the report or the decision to submit the report for publication.

**Competing interests.** The authors declare none.

**Ethics statement.** The authors assert that all procedures contributing to this work comply with the ethical standards of the relevant national and institutional committees on human experimentation and with the Helsinki Declaration of 1975, as revised in 2008.

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
