## [Reviewer Report]

This study sheds light on the preferences and experiences of Syrian MHPSS users in NW Syria and Türkiye by a qualitative analysis of structured interviews with 378 displaced Syrians.

The methodology is sound, the number of participants more than adequate and the manuscript is very well written. The research findings and the complementing narratives of service user experiences and preferences, are valuable both to improve the quality of direct service user care as well as to guide the rebuilding of the (mental) health system.

Minor comments:

1. Title: notice that “mental health and social services” is used and not “mental health and psychosocial services (MHPSS)” as used in the rest of the text.

2. Study design: it remains unclear how the 9 participating organisations were selected and if Hope Revival Organization was amongst them. If so, how did the researchers control for potential bias in the service users responses in view of the continuing dependence on the services?

3. Data analysis:

• Readers would benefit from a brief explanation on the adaptations made to the WHO framework mentioned and adopted.

• The authors managed to attract a high number of participants (n=378). This number gives every reason to assume that the point of data saturation was reached, but it would add strength to the manuscript if the authors briefly mention this point.

4. Discussion:

• I would suggest to put the 8 elements in order of importance given (indicated by frequency mentioned) by the participants.

• Did the qualitative data from the interviews give rise to any hypothesis on the reason for the large discrepancies between men and women in the elements of confidentiality and continuity of care?

• The authors make a case for localisation of services, using local service providers with lived experience. While the authors’ arguments for this are strong, the discussion would benefit from the inclusion of a critical note too. For instance a consideration on the high risks of secondary traumatization in this already vulnerable group. Or on the need for training on knowledge, skills and attitudes in order to professionally embed one’s lived experience.

5. Limitations: did the authors take into account the possible impact of the use of telephone interviews rather than face-to-face interviews on the response rate and content?

Writing errors:

1. Page 6, line 44: During a process of data familiarisation and discussion of within the team …

2. Page 9, line 57: Service users valued being actively involved…

3. Page 14, line 6: … in other health settings, including and non-communicable diseases.

4. Page 16, line 19: … has been undermined by years of exclusion and …

---

## [Reviewer Report]

This is an original study about the preferences of Syrian refugees relative to mental health care, based on a large sample of 378 participants. The study is quite thorough, and my questions have to do with clarifications rather than major comments:

1. the time of data collection and the methods used to analyse the study should be described in detailed

2. the issue of being able to access treatment in one’s mother language is not discussed - isn’t that one of the essential points to be able to establish trust?

3. discussion of participants‘ mental health in the context of their everyday living conditions should be broader - to what extent is participants’ preference influenced by their mental health, their living circumstances, age, gender?

4. what is the role of past experiences with the mental health care system with regard to individuals’s preferences?

5. finally, in what way are the Syrian patients interviewed in this project different from patients from other geographical areas?

---

## [Editor Report]

This manuscript presents an original and well-executed qualitative study exploring the preferences and experiences of Syrian MHPSS users in northwest Syria and Türkiye, based on structured interviews with 378 displaced Syrians. Reviewers praised the methodological rigor, clarity of writing, and the practical relevance of the findings for improving service quality and informing mental health system strengthening in conflict-affected contexts. As detailed in their specific comments, they have requested some points of clarification and some additional reflections within the discussion.

---

## [Editor Report]

Thank you for submitting the revised version of your manuscript, which sufficiently addressed all reviewer concerns. I agree with the reviewer recommendation to publish the paper. Congratulations!